# Impact of change in household environment condition on morbidity in India: Evidence from longitudinal data

N. Brahmanandam *, R. Nagarajan

Department of Development Studies, International Institute for Population Sciences (IIPS), Mumbai, Maharashtra, India

* brahmameco@gmail.com

## Abstract

### Background

Household environment condition is an important predictor of morbidity of the household members. Without forming a healthy household environment, creating a healthy population is not possible. In this background, this study assesses the impact of change in household environment conditions on morbidity.

### Methods

For the empirical analysis purpose of this study, we used two waves of longitudinal data from India Human Development Survey (2004–05, 2011–12). This study is based on 34131 re-contacted households in 2011–12 from the base year 2005. The bivariate and ANOVA tests were performed to assess any short-term morbidity (diarrhoea, fever and cough) with respect to change in household environment condition from 2005 to 2011. The multivariate linear regression was performed to assess the impact of change in household environment conditions on morbidity. The multinomial logistic regression was used to assess the impact of change in household environment condition on change in morbidity.

### Results

The results from multivariate linear regression have shown that the share of household members fell sick due to any short-term morbidity (ASM) was significantly lower (β = −0.060, P<0.001) among the households who lived in clean environment condition in both the periods, 2004–05 and in 2011–12 as compared to those who were living in poor environment condition in both periods net of other socio-economic characteristics of the households. The share of household members fell sick due to any short-term morbidity has significantly declined (β = −0.051, P<0.001) among the household whose household environment condition has changed from poor in 2004–05 to clean environment in 2011–12 as compared to the households who have lived in poor environment condition in both periods in 2004–05 and 2011–12. The results of adjusted percentage from multinomial logistic regression have shown that the household members who fell sick with ASM was remained higher

**Data Availability Statement:** The data used in this study is available in public domain https://www.icpsr.umich.edu/web/DSDR/studies/37382/datadocumentation

**Funding:** The author(s) received no specific funding for this work.

**Competing interests:** The authors have declared that no competing interests exist.

(4.9%; P<0.05) among the households whose environment condition was remained poor in both years in 2005 and 2011 as compared to the other households (2.7%) who remained in the better-off condition in both years in 2005 and 2011.

## Conclusion

Considering the findings of the study, we suggest that ongoing government flagships programmes such as *Swacch Bharat Mission* (Clean India Mission), *Pradhan Mantri Ujjwala Yojana* (Prime Minister Clean Energy Scheme) and *Pradhan Mantri Awas Yojana* (Prime Minister Housing Scheme), and *Jal Jeevan Mission* (Improved Source of Drinking Water Scheme) should work in tandem to improve household environment conditions.

## Introduction

Household environment condition has an important place in public health policy. A healthy household environment condition makes healthy people. The evidence from developing countries suggests that lack of safe drinking water, improved sanitation facilities, hygienic conditions, and improved cooking fuel are associated with morbidity and mortality [1–7]. Similarly, studies have also found that diarrheal disease and measles infections are linked to a shortage of water, water contamination, household crowding, and poor housing [8–11]. The poor water, sanitation, and hygiene account for 4% of all deaths and 5.7% of disability-adjusted life years globally [12]. Also, the poor household environment conditions are the second major cause of death among the children [13].

Because a large percentage of diarrheal morbidities are attributable to mortality due to lack of safe drinking water, improved sanitation, and insufficient hygiene, at the end of the Millennium Development Goals (MDGs), the policies emphasised and developed to reduce the percentage of people living in the households with lack of sustainable environment in terms of durable housing structure, access to safe water, and improved sanitation [14]. After the MDGs transition, the Sustainable Development Goals (SDGs) emphasized household environmental protection at the center of discussion with goals such as universal access to safe drinking water, basic sanitation, and affordable clean energy for all [15].

Akin to the global policy shifts, India has initiated several policies and programme interventions such as National Rural Drinking Water Programme, Total Sanitation Programme, Household Clean Energy Programme, and National Rural Health Mission during the period 2005–10. These policies have emphasised household environment condition and made targets to achieve WASH (water, sanitation and health) facilities. Due to poor operational management of the above programmes, India achieved little progress in WASH indicators such as improved drinking water source and modern/improved-sanitation facilities [16]. The poor household environment condition is still prevailing. The main goal of MDG 7, target 7c was to reduce the proportion of households without access to improved sanitation from 51% in 1991 to 25% in 2015 [17]. India has not met this target, though a little progress has been achieved in improved sanitation facilities. Only 49% of the households were using improved toilet facilities in India in 2015–16 and 39% of the households were defecating openly. The percentage of household using improved cooking fuel was 44% in India in 2015–16 [18]. Thus, the level of poor household environment conditions remains a problem for a sustainable and inclusive society. In this context, we attempted to assess the impact of change in household environment conditions on morbidity in India, for the first time, using longitudinal data. The study

contributes to the literature by providing the causal relationship between the change in the household environment and morbidity among the household members.

## Background

A primarily reported component of the household environment is the availability and accessibility of water, sanitation, hygiene, and cooking fuel. Despite India's remarkable economic growth during 2005 and 2011, many households are still struggling to meet their most basic needs, including availability and access to good housing conditions, water and sanitation, and clean energy for cooking [19, 20]. The World Bank estimates reveal that India specifically loses 6.4% of its gross domestic product (GDP) every year for water and sanitation-related diseases [21]. Sanitation-related diseases cause loss of time for normal activities for adults as well as children. The loss of time due to more illness can lead to substantial losses in productivity, welfare, income, and lifetime opportunities for patients and for family members who care for them. Also, the estimates reveal that the overall treatment cost contributed by sanitation and hygiene-related diseases is 38% [21]. Similarly, improving the household environment condition improves health by freeing people from illness and also increase the productivity of the people resulting in economic gain [22]. Improvement in household environment conditions such as water, sanitation, hygiene, and cooking fuel reduces stunting and increases children's height [23, 24] much more than the direct nutritional interventions [25].

### Effect of sanitation and hygiene on morbidity

Human waste is the reservoir for a range of pathogenic bacteria and soil-transmitted helminths which cause diarrhea and other morbidities among young children [26, 24]. Thus, open defecation and inadequate sanitation facilities are closely associated with infectious diseases. Studies observed that people in rural India prepare food and touch babies' mouths with hands contaminated by the child or adult feces which reflects poor hygienic practices. Moreover, open defecation is not limited to a remote field. In rural India, it was often observed that children were defecating in and near home and playing near open defecation areas [25]. Most of the studies found that the higher mortality in developing countries is due to disease associated with poor sanitation and personal and household hygienic condition [26]

### Effect of water on morbidity

Access to water supply and sanitation is a fundamental need and human right. It is vital for the dignity and health of all people [28, 29]. The water intended for human consumption should be safe and wholesome, which is defined as (a) free from pathogenic agents, (b) free from harmful chemical substances, (c) pleasant tasty, for instance free from colour and odour, and (d) usable for domestic purposes. The water is said to be polluted and contaminated when it does not fulfill the above criteria. Nevertheless, 29 percent of the world population lack safe drinking water and around 61 percentage lack sanitation facilities [15]. It is evident from India that the use of contaminated water causes morbidities like diarrhea, fever and jaundice [30].

### Effect of cooking fuel on morbidity

Previous studies have indicated that biomass combustion for cooking such as wood, cow dung, crop residues, and charcoal in indoor releases respiratory irritants such as particulate matter (PM), carbon monoxide (CO), sulfur dioxide, NO2, and organic toxins. These air pollutants have been associated with increased symptoms of asthma [31–37]. All these air pollutants cause respiratory and infectious diseases such as Acute Respiratory Infection (ARI),

Asthma, Tuberculosis (TB), and eye diseases [38, 39]. The children are at a greater risk of developing ARI and women and elderly have a greater risk of developing Asthma and TB due to exposure to indoor air population [30].

The previous studies have largely investigated the association between household environmental components such as water, sanitation, hygiene, and cooking fuel with morbidities such as diarrhea and respiratory diseases. To our knowledge to date this is the first study investigate the causal relation between any short term morbidity (Diarrhea, fever and cough) and household environment condition in the context of India using data from the survey that followed the same households in 2004–05 and 2011–12.

## Materials and methods

### Data source

The present study uses data from two rounds of a longitudinal survey of IHDS 2004–05 and 2011–12. It is a nationally representative survey carried out in collaboration with the researchers at the University of Maryland and the National Council of Applied Economic Research (NCAER), New Delhi. The First (IHDS-I 2004–05) wave of the survey was administered to a nationally representative sample of 41,554 households. IHDS-II (2011–12) re-interviewed 83 percent of households from IHDS-I, with the additional replacement of the sample of 2,134 households [40]. The total sample size of IHDS-II was 42,152 households. Both surveys have collected household information on income, education, health, consumption expenditure, fertility, and family planning. These surveys also collected information on household environment conditions. Both surveys covered all the states and union territories of India except Andaman/Nicobar and Lakshadweep Islands. For analytical purpose of this study, we used the sample size of 34,131 households in 2011–12 which has been followed-up from the base period of 2004–05.

### Sample design and study population

The IHDS households are spread across 33 states and union territories, 384 districts, 1,503 villages, and 971 urban blocks, located in 276 towns and cities. Villages and urban blocks (comprising of 150–200 households) formed the primary sampling unit (PSU) from which the households were selected. Urban and rural PSUs were selected using different designs. To draw a random sample of urban households, all urban areas in a state were listed in the order of their size with the number of blocks drawn from each urban area allocated based on probability proportional to size. Once the numbers of blocks for each urban area were determined, the enumeration blocks were selected randomly with help from the Office of the Registrar General of India. From these Census Enumeration Blocks of about 150–200 households, a complete household listing was conducted and household samples of 15 households per block were selected [41].

### Variables

There are two outcome variables in this study. The first outcome variable is a linear variable: the percentage of household members fell sick due to any short-term morbidity during past one month preceding the survey period. The percentage share of household members fell sick due to any short-term morbidity was measured at household level. The second outcome variable of any short-term morbidity is binary nature in two rounds of survey and it was converted into multinomial variables comprising four categories indicating the transitional nature of any short-term morbidity among household members: (a) the percentage of household members fall sick due to any short term morbidity (hereafter ASM) remains the same in 2005 and 2011; (b) the percentage of household members fall sick due to ASM in 2005 but is not sick in 2011. 3) the percentage

of household members who were not sick in 2005 but fell sick due to any ASM in 2012; (d) the percentage of household members remain without ASM in 2005 and 2011.

The main predictor variable in this study is the change in household environment conditions (hereafter HEC). It is classified into four categories to reflect the transition pattern: (a) Remaining in poor HEC in 2005 and 2011; (b) Poor HEC in 2005 and non-poor HEC in 2011; (c) Non-poor HEC in 2005 and became poor in HEC in 2011; (d) Remaining in non-poor HEC in 2005 and 2011. The variables controlled are various socio-economic indicators such as place of residence (rural and urban), type of house (*kachha* and *pucca*), occupation of the head of the household (primary, secondary, tertiary, and no-occupation), educational level of the head of the household (illiterate, primary, secondary, and higher education), economic status of the household (poor and non-poor), social group category (General, OBC, SC, and ST), religious groups (Hindu, Muslim, Christian, and Others), and regions (North, Central, East, Northeast, West, and South). The descriptive statistics of all these variables and sample size for round-1 and round-2 of IHDS are described in Table 1.

## Statistical approach

By using the information on HEC, 18 variables were dichotomised as 0 for disadvantaged and 1 for the advantaged group for both periods, 2004–05 and 2011–12 (Table 2). The HEC index was constructed by using Principal Component Analysis (PCA). For each of the HEC variable, a weight (factor score) generated from the PCA was assigned. The result of HEC scores was standardized in relation to a normal distribution with zero mean and standard deviation one, then values divided into three equal parts as poor, middle, and better off condition of households for two periods. The estimation of the HEC index was adopted from DHS methodology [27, 43]. The reliability, validity, and suitability of 18 components used in the index were carefully examined both quantitatively and theoretically. The reliability coefficients are reported at the bottom of Table 1. As pointed out above, the index was classified into four categories to represent the transition pattern: (a) remaining in poor HEC in 2005 and 2011; (b) poor HEC in 2005 and non-poor HEC in 2011; (c) non-poor HEC in 2005 and became poor in HEC in 2011; (d) remaining in non-poor HEC in 2005 and 2011.

The analyses were carried-out in two stages. In the first stage, we modeled bivariate analyses and an ANOVA test to estimate the significant differences in prevalence of ASM among household members by the change in HEC across the various socio-economic characteristics of households. In the second stage, we have modelled two sets of analyses based on the nature of dependent variables. First, for continuous dependent variables, we have used the multivariate linear regression to assess the impact of change in HEC on the percentage of household members who fell sick due to ASM. Second, we applied the multinomial logistic regression and Multiple Classification Analysis (MCA) conversion model to see the impact of change in HEC in response to change in morbidity condition of the household members by controlling for various socioeconomic characteristics.

Multinomial Logistic Regression and MCA conversion model: We used MCA conversion model to estimate adjusted percentage of transition in household members fell sick due to ASM by change in HEC and other controlled socioeconomic variables.

The mathematical equation of MCA multinomial analysis is as follows:

$$Z_1 = Log\left(\frac{P_1}{P_4}\right) = a_1 + \sum b_{1j} * X_j$$

**Table 1. Description of percentage and sample size by background characteristics followed in IHDS-1 2004–05 and IHDS-2 2011–12.**

| | 2004–05% | N | 2011–12% | N |
|---|---|---|---|---|
| **Place of residence** | | | | |
| Rural | 75.4 | 23,642 | 72.5 | 22,841 |
| Urban | 24.6 | 10,489 | 27.5 | 11,290 |
| **Type of house** | | | | |
| Kucha | 67.71 | 21,505 | 59.7 | 18,820 |
| Pucca | 32.29 | 12,626 | 40.3 | 15,311 |
| **Occupation of household head** | | | | |
| Primary | 66.1 | 21,829 | 64.7 | 21,270 |
| Secondary | 9.6 | 3,292 | 18.8 | 6,466 |
| Tertiary | 16.6 | 6,166 | 15.4 | 6,017 |
| No-occupation | 7.7 | 2,844 | 1.0 | 378 |
| **Education of household head** | | | | |
| Illiterate | 63.9 | 21,541 | 62.2 | 21,100 |
| Primary | 13.0 | 4,331 | 11.9 | 4,004 |
| Secondary | 20.4 | 7,194 | 22.3 | 7,737 |
| Higher | 2.8 | 1,065 | 3.6 | 1,290 |
| **Economic status** | | | | |
| Poor | 22.1 | 7,154 | 18.4 | 5,768 |
| Non-poor | 77.9 | 26,977 | 81.6 | 28,363 |
| **Caste** | | | | |
| General | 28.0 | 10,546 | 25.9 | 9,732 |
| OBC | 41.5 | 13,564 | 41.8 | 13,713 |
| SC | 22.8 | 7,157 | 22.7 | 7,192 |
| ST | 7.8 | 2,864 | 8.0 | 2,961 |
| **Religion** | | | | |
| Hindu | 83.0 | 27,736 | 83.2 | 27,942 |
| Muslim | 11.0 | 3,822 | 11.1 | 3,850 |
| Christian | 6.0 | 2,573 | 2.4 | 1,059 |
| Others | | | 3.3 | 1,280 |
| **Region** | | | | |
| North | 13.4 | - | - | 7,624 |
| Central | 21.5 | - | - | 6,584 |
| East | 22.9 | - | - | 5,807 |
| Northeast | 3.1 | - | - | 1,422 |
| West | 15.6 | - | - | 4,683 |
| South | 23.5 | - | - | 8,011 |
| **Change in HEC** | | | | |
| Poor HEC in 2005 & 2011 | 26.3 | - | - | 7,442 |
| Poor HEC in 2005 but Non-poor in 2011 | 12.1 | - | - | 3,703 |
| Non-poor HEC in 2005 but Poor in 2011 | 11.5 | - | - | 3,504 |
| Non-poor HEC in 2005 & 2011 | 50.2 | - | - | 19,482 |
| **Change in ASM** | | | | |
| No-sick 2005 & 2011 | 73.2 | - | - | 24,993 |
| No-sick 2005 & sick 2011 | 13.8 | - | - | 4,710 |
| Sick 2005 & no-sick 2011 | 9.74 | - | - | 3,326 |
| Sick 2005 & 2011 | 3.2 | - | - | 1,102 |
| **Mean percent ASM** | 11.9 | 34,131 | 18.1 | 34,131 |

**Note: Change in HEC** stands for Change in Household Environment Condition; **Change in ASM** stands for change in Any Short-term Morbidity; **N** stands for sample size.

**Table 2. Indicators included in household environment condition index.**

1. Households with improved source of drinking water are coded as 1 and it is 0 otherwise. The improved source of drinking water includes piped water, tube well, hand pump, covered well, rain water and bottled water. While unimproved source of drinking water includes open well, river, pond, truck and others.

2. Households with supply of water for more than one hour is considered as better availability (1) compared to less than one hour (0)

3. Households having vessel with lid for drinking water storage are considered hygienic (1), while no vessel for storage and vessels with no lid are unhygienic (0)

4. The households always purify the drinking water have better hygienic conditions (1) than those which never, rarely, sometimes and usually purify (0).

5. Improved methods of pouring drinking water are using long ladle and tap in the vessels (1), while the unimproved methods include cups and utensils (0)

6. The household's members spend time less than or equal to 30 minutes to fetch water per day have better accessibility to water (1) than those who spend more than 30 minutes (0).

7. The households having improved toilet facility as traditional latrine, VIP latrine, and flush toilet (1), while non-availability of toilet facility is open defecation (0).

8. Better accessibility of toilet facilities comprises toilets within the dwelling, shared toilet inside and outside building, and public toilets (1). Not accessibility of toilet facility is open defecation (0).

9. The members of the households practicing hand wash after defecation is grouped as hygienic practices (1) and it is unhygienic on the other way (0).

10. Members of the households wash hands using soap after defecation are grouped into hygienic practice (1), while the hand wash using other material such as water alone, mud/ash and others are unhygienic practices (0).

11. The households with separate kitchen are categorized as having improved cooking place (1), and unimproved conditions include cooking in outdoors and in living area (0)

12. The household having availability of ventilation in cocking place is considered (1). while there is no ventilation and cooking in outdoor is (0)

13. The households use improved (1) and unimproved (0) cooking fuels. The improved fuels are improved fuels are LPG and kerosene and unimproved are firewood cow dung, crop residue, coal and charcoal

14. Households having electricity are considered (1) while there is no electricity (0)

15. The households having improved building materials viz. improved wall, roof and floor are considered as pukka house (1), and the houses are kutcha otherwise (0). The improved materials of wall are burnt bricks, stone and cement/concrete, while the improved includes grass, thatch, mud, unburnt bricks, plastic, woods, GI/metal sheets and others. Similarly, the improved roof materials are cement/concrete, bricks and stones, while the improved materials comprises grass, mud, wood, tiles, slate, plastic, GI/metal sheets, asbestos and others. The improved floor types incorporate cement/concrete and tiles/mosaic, while the improved materials are mud, unburnt bricks, wood, bamboo, bricks, stone and others

16. The households with better housing space are those which have three and less persons per sleeping room (1) as compared to more than two persons per sleeping room (0).

17. The households having stagnant water on surrounding area of the house is considered (0) otherwise no stagnant of water (1)

18. The households having human or animal excrement surrounding house is considered (0), while there is no human or animal excrement surrounding house (1)

Reliability coefficients: Estimated Household Environment Condition Index's Alpha coefficient in 2004–05 is 0.73 and in 2011–12 is 0.75 which shows reliable and consistency of 18 variable used in Index.

$$Z_2 = Log\left(\frac{P_2}{P_4}\right) = a_2 + \sum b_{2j} * X_j$$

$$Z_3 = Log\left(\frac{P_3}{P_4}\right) = a_3 + \sum b_{3j} * X_j$$

$$\text{And } P_1 + P_2 + P_3 + P_4 = 1$$

Where,

ai i = 1,2: constants

bij i = 1,2; j = 1,2. . ..n: multinomial regression coefficient.

$P_1$ = Estimated probability of household members fell sick due to ASM remained the same in 2005 and 2011.

$P_2$ = Estimated probability of household members not sick due to ASM in 2005 but fell sick due to ASM in 2011.

$P_3$ = Estimated probability of household members fell sick due to ASM in 2005 but not sick due to ASM in 2011.

$P_4$ = Estimated probability of household members remained not sick due to ASM in 2005 and 2011.

Hear $P_4$ is a reference category

For the sake of simplicity in the interpretation of results, multinomial logistic regression coefficients were converted into adjusted percentages. The procedure consists of following steps:

Step 1:

By using regression coefficient and mean values of independent variables, the probability was computed as:

$P_i = \frac{\exp{(Z_i)}}{\left\{1 + \sum \exp{(Z_i)}\right\}}$, i = 1, 2, 3, 4 and $P_4$ = 1-$P_1$ + $P_2$ + $P_3$ where Z was the estimated value of response for all categories of each variable.

Step 2

To obtain the percentage values, the probability P was multiplied by 100.

[For mathematical proof of regression model refer to Retherford and Choe [44]]. All the analyses for this paper were carried out by using STATA 13.1 version.

## Ethical statement

The current study has used secondary data source from the two rounds of India Human Development Survey, 2004–05 and 2011–12. The ethic review board at the University of Maryland, U.S.A and the National Council of Applied Economic Research (NCAER), New Delhi, India granted IHDS-1 and IHDS-2 project ethical approval before surveys were conducted, with written informed consent obtained from survey participants during the survey. This survey was also reviewed and approved by ICF International Review Board (IRB). IHDS-1 and IHDS-2 surveys are anonymous public available data set with no identifiable information of survey participants. The data is available in public domain and can be accessed from IHDS website at https://www.icpsr.umich.edu/web/DSDR/studies/37382/datadocumentation. Therefore, no separate ethical approval is required for this study.

## Results

### Change in HEC and ASM in different socioeconomic characteristics

Table 3 shows that the average percentage of members of households fell sick due to ASM (diarrhea, fever, and cough) by the change in HEC with various socio-economic characteristics in India. The results indicate that the average percentage of household members fell sick due to ASM significantly varies across all socioeconomic characteristics of the household.

**Table 3. Average percentage of household members fell sick due to ASM by change in household environment condition and socio-economic characteristics of the households.**

| Background variables | Any short term morbidity by change in Household environment condition | | | | |
|---|---|---|---|---|---|
| | Poor HEC in 2005 & 2011 | Poor HEC in 2005 but Non-poor HEC in 2011 | Non-poor HEC in 2005 but Poor HEC in 2011 | Non-poor HEC in 2005 & 2011 | Anova test value and significance level |
| **Place of residence** | | | | | |
| Rural | 21.9 | 18.3 | 22.6 | 16.8 | 14.97*** |
| Urban | 15.7 | 15.0 | 15.3 | 13.5 | |
| **Occupation of household head** | | | | | |
| Primary | 21.3 | 18.3 | 22.8 | 15.9 | 9.75*** |
| Secondary | 20.9 | 15.5 | 17.7 | 15.8 | |
| Tertiary | 23.9 | 14.0 | 19.7 | 14.4 | |
| No-Occupation | 22.3 | 21.9 | 18.7 | 14.5 | |
| **Education of household head** | | | | | |
| Illiterate | 21.6 | 18.3 | 22.4 | 16.2 | 11.11*** |
| Primary | 19.7 | 15.5 | 22.0 | 15.2 | |
| Secondary | 23.4 | 18.1 | 18.1 | 14.4 | |
| Higher | 20.8 | 23.2 | 31.4 | 14.2 | |
| **Economic status** | | | | | |
| Poor | 20.7 | 20.2 | 23.6 | 16.0 | 28.09*** |
| Non-poor | 22.1 | 17.1 | 21.1 | 15.4 | |
| **Caste** | | | | | |
| General | 23.1 | 18.2 | 20.3 | 14.8 | 12.22*** |
| OBC | 24.0 | 17.9 | 23.1 | 15.8 | |
| SC | 21.4 | 18.7 | 22.3 | 17.2 | |
| ST | 14.0 | 14.3 | 16.5 | 10.8 | |
| **Religion** | | | | | |
| Hindu | 21.8 | 17.6 | 22.2 | 15.5 | 20.23*** |
| Muslim | 24.7 | 20.0 | 19.9 | 15.2 | |
| Christian | 10.8 | 16.4 | 14.5 | 15.1 | |
| **Region** | | | | | |
| North | 18.3 | 20.6 | 19.5 | 17.0 | 27.73*** |
| Central | 27.9 | 26.7 | 32.2 | 23.1 | |
| East | 19.3 | 18.3 | 18.6 | 18.2 | |
| Northeast | 16.2 | 10.5 | 10.4 | 10.8 | |
| West | 13.1 | 11.8 | 15.6 | 10.8 | |
| South | 17.3 | 16.5 | 17.2 | 13.2 | |
| **Anova test value and significance level** | | | | | |
| **Total** | **21.6** | **17.9** | **21.8** | **15.4** | |

*** stand for P<0.01.

Specifically, the results indicate that the average percentage of household members fell sick due to ASM was higher among the households that lived in poor HEC in both 2005 and 2011 (21.6%) as compared to households who remained non-poor in HEC in both 2005 and 2011 (15.4%) in India. The average percentage of household members fell sick due to ASM has declined (17.9%) among the households that have experienced a better change in HEC in 2011 as compared to 2005. Whereas, average percentage of household members fell sick due to morbidity has increased (21.8%) among the households that have lived in a clean HEC in 2005 but moved into poor household environment conditions in 2011. This indicates that the

household members who are living in a clean environment condition not only having less morbidity but also healthier as compared to those household members living in non-clean environment.

With respect to the economic status (poor vs non-poor) of the households, among the household members who have lived in poor HECs in both 2005 and 2011, the average percentage of morbidity of poor and non-poor household members was higher (20.7% and 22.1% respectively) compared with the poor and non-poor household members who have lived in clean HECs in 2005 and 2011 (16% and 15.4% respectively). Among the economically poor and non-poor households who have lived in poor HEC in 2005 but they turned into clean environment condition in 2011, their morbidity has declined (20.2% and 17.1% respectively). Whereas, economically poor and non-poor household members who have lived in clean environment condition in 2005 but they turned into poor environment condition in 2011, their morbidity has increased (23.6% and 21.1% respectively). It has been observed that even among the economically poor as well as non-poor members of households who continued living in a clean environment condition has shown less morbidity compared to those living in worse environment condition. The household environment condition appears to be an important determinant of health of household members. It is also found that the average percentage of household members fell sick due to ASM declined among the households whose HEC has changed from poor in 2005 to non-poor HEC in 2011. The morbidity has increased among the households whose environment condition became worse in 2011 from clean in 2005, irrespective of economic status.

The similar pattern was observed across rural and urban households, educational categories of household head (illiterate, primary, secondary and higher), caste groups (General, OBC, SC and ST), religious categories (Hindu, Muslim and Christian) and regions (North, South, East, West, Northeast and Central) in India with respective change in HEC. The ANOVA test has shown significance for the average percentage of household members fell sick due to ASM with respect to change in HEC across various social and economic characteristics of the households.

## Change in ASM in relation to change in HEC across states

Table 4 shows state-wise average percentage of household members fell sick due to any short term morbidity with respect to change in HECs in India. It is observed that the morbidity prevalence of household members in all the states have shown decrease when their HEC has transformed from poor in 2005 to non-poor in 2011, In contrast, the morbidity prevalence of household members in all the states have shown increase when their HEC has turned into poor in 2011 form non-poor in 2005. The results has also shown that, when the HEC remained in improved status in both periods, the percentage of morbid members in the household is lower in comparison to those households whose environment condition was remained in poor for both periods across the states.

The household members who were living in poor HEC in both periods, the prevalence of household members fell sick due to ASM is highest for Punjab (35.5%), followed by Chhattisgarh (29.5%), Uttar Pradesh (29.2%), Bihar (28.3%) and Uttaranchal (28.2%). The top five states having the highest percentage of morbid population where the HEC has worsened from non-poor in 2005 to poor in 2011 are Uttar Pradesh (35.8%), Uttaranchal (26%), Jammu & Kashmir (25.5%), and Andhra Pradesh (25.2%).

The socio-economically underdeveloped states such as Chhattisgarh (29.5%), Uttar Pradesh (29.2%), Bihar (28.3%), and Uttaranchal (28.2%) and economically better-off state Punjab (35.5%), have shown higher morbidity with respect to poor HEC. The poorer states may be

**Table 4. Average percentage of members of households fell in sick in any short term morbidity by change in household environment condition by states.**

| States | Average percentage of short term morbidity by change in household environment condition | | | |
|---|---|---|---|---|
| | Poor HEC in 2005 & 2011 | Poor HEC in 2005 but Non-poor HEC in 2011 | Non-poor HEC in 2005 but Poor HEC in 2011 | Non-poor HEC in 2005 & 2011 |
| Jammu & Kashmir | 18.8 | 13.3 | 25.5 | 11.0 |
| Himachal Pradesh | 21.6 | 17.7 | 21.0 | 19.4 |
| Punjab | 35.5 | 32.5 | 23.1 | 20.8 |
| Uttaranchal | 28.2 | 44.2 | 26.0 | 24.6 |
| Haryana | 18.1 | 12.1 | 18.6 | 13.7 |
| Delhi | 15.7 | 7.9 | 10.5 | 11.7 |
| Rajasthan | 17.1 | 14.4 | 17.8 | 14.6 |
| Uttar Pradesh | 29.2 | 26.9 | 35.8 | 25.5 |
| Bihar | 28.3 | 28.2 | 20.4 | 18.0 |
| Tripura | 9.6 | 11.5 | 19.2 | 7.2 |
| Assam | 16.8 | 11.1 | 10.6 | 14.4 |
| West Bengal | 19.9 | 19.9 | 21.4 | 20.7 |
| Jharkhand | 11.7 | 10.7 | 13.2 | 11.5 |
| Odisha | 11.8 | 8.4 | 16.3 | 13.1 |
| Chhattisgarh | 29.5 | 29.2 | 34.8 | 24.4 |
| Madhya Pradesh | 23.2 | 23.3 | 22.5 | 16.0 |
| Gujarat | 11.8 | 10.9 | 13.0 | 11.0 |
| Maharashtra | 14.0 | 12.0 | 16.9 | 10.7 |
| Andhra Pradesh | 21.4 | 20.4 | 25.2 | 18.8 |
| Karnataka | 12.2 | 12.2 | 14.1 | 11.0 |
| Tamil Nadu | 16.1 | 39.0 | 13.2 | 9.5 |
| Pondicherry | 15.8 | 12.6 | 0.0 | 2.0 |
| Kerala | 0.0 | 39.0 | 23.1 | 14.0 |
| Total | 21.6 | 17.9 | 21.8 | 15.4 |

less conscious of their HEC and that may not have sufficient WASH facilities. The economically better-off state Punjab is shown higher morbidity (35.5%) with respect poor HEC due to higher utilization of health care service. There are some states such as Uttaranchal (44.2%), Kerala (39%), Punjab (32.5%), Chhattisgarh (29.2%), and Bihar (28.2%) which have shown exceptionally higher morbidity with respect to improved HEC from poor in 2005 to non-poor in 2011. The states like Punjab, Uttaranchal and Kerala have shown a higher percentage of morbidity than the states with poor HEC. This may be due to the increase in utilization of health care services from 2005 to 2011. Higher morbidity in Kerala may be attributed to the increase in the share of old age population as the states in in advanced stage of demographic transition.

## Transition in ASM with respect to the transition of HEC in India

Table 5 indicates the change in ASM in response to changes in the household environment condition in India. The prevalence of morbidity is observed to be low in the households which either have the better-off environment condition in both years (2.8%) or shifted to better-off environment condition in the recent year (4.2%) as compared to their counterparts who were

**Table 5. Transition in percentage of any short term morbidity of members of households by change in household environment condition, IHDS 2005 and 2011.**

| Change in household environment condition in India | Change in any short term morbidity | | | |
|---|---|---|---|---|
| | No ASM in 2005 & 2011 | No ASM in 2005 & ASM in 2011 | ASM in 2005 & No ASM in 2011 | ASM in 2005 & 2011 |
| Poor HEC in 2005 & 2011 | 66.1 | 16.6 | 11.9 | 5.4 |
| Poor HEC in 2005 and non-poor HEC in 2011 | 69.8 | 15.1 | 10.9 | 4.2 |
| Non poor HEC in 2005 and poor HEC in 2011 | 67.5 | 16.7 | 11.4 | 4.5 |
| Non-poor HEC in 2005 & 2011 | 75.4 | 12.9 | 8.9 | 2.8 |
| Total | 71.4 | 14.6 | 10.2 | 3.8 |

Pearson chi2(9) = 218.2871, Pr < 0.0001

remained in the poor environment condition either in both years 2005 and 2011 (5.4%) or became worse-off in the environment condition in recent year (5.4%). The percentage of household members who were not sick in 2005 and fell in sick in 2011 was low in the households either have better-off living environment in both years (12.9%) or improved to better–off in the recent year (15.1%) as compared to their counterparts who were remained in the poor living environment either in both years (16.6%) or became worse-off in the living environment in the recent year (16.7%). The chi square statistics has shown the significant association between the changes in percentage of household members fell in sick due to any short-term morbidity and HEC. (Pearson chi2(9) = 218.1, Pr < 0.0001).

## Impact of change in HEC on ASM

Fig 1 shows that the normal probability curve of household members fell sick due to ASM. The dependent variable, the percentage of household members fell sick due to ASM, is log-transformed and normally distributed. Therefore, we can perform the multivariate linear regression for our analysis.

Table 6 presents the association between members of households fell sick due to ASM and the change in HEC by socio-economic variables. The results of multivariate linear regression show that the percentage of household members fell sick due to ASM is significantly lower (β = −0.060, P<0.001) among the households that have lived in clean household environment in 2005 and 2011 as compared to the households who were remained in poor living* environment condition in both periods after controlling for other socio-economic characteristic of the households. The percentage of household members fell sick due to ASM is significantly lower (β = −0.051, P<0.001) among the households whose environment condition has been changed from poor in 2005 to non-poor in 2011 as compared to the households who were remained in poor environment condition in both periods. The percentage of household members fell sick due to ASM is significantly higher (β = 0.089, P<0.001) among the non-poor households as compared to the poor households. This is may be due to the higher reporting and utilization of health care services among non-poor households. Among religious groups, Muslims are less likely to get morbid (β = -0.104, P<0.001) as compared to the Hindus. The morbidity is higher in Central (β = 0.147, P<0.001), Eastern (β = 0.053, P<0.001) and Southern region (β = 0.118, P<0.001) as compared to Northern region. Morbidity in Western part of India is significantly lower (β = -0.054, P<0.001) in comparison with northern India.

Table 7 shows the impact of change in HEC and demographic and socio-economic factors on likelihood reporting of a change in ASM by fitting the multinomial logistic regression model. The adjusted percentage of household members fell sick with the change in ASM by

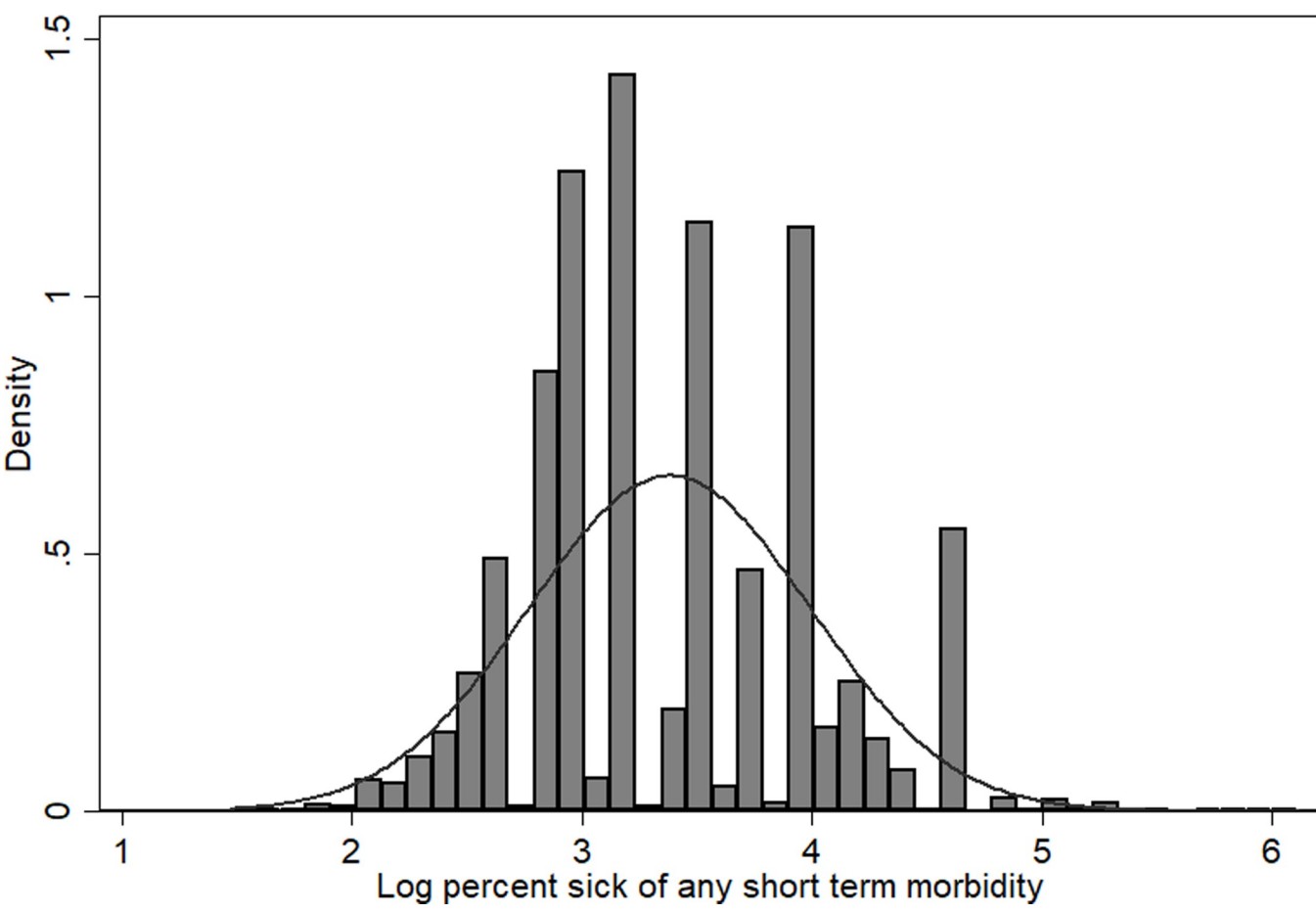

**Fig 1. Normal probability curve for percentage share of household members fall sick due to any short term morbidity.**

the change in HEC and demographic and socio-economic factors are arranged. The significant differences were observed between change in percentage of household members fell sick due to ASM and change in HEC. The household members who fall sick with ASM was remained higher (4.9%; P<0.05) among the households whose environment condition was remained poor in both years in 2005 and 2011 as compared to the other households (2.7%) who remained in better-off condition in both years. The members of household who fell sick with ASM remained higher (3.9%; P<0.01) among the households whose environment has been turned downwards into poor condition in 2011 from better-off condition in 2005 as compared to the other households (2.7%) who remained in better-off condition in both years. While the members of households who were not sick in 2005 and fell in sick in 2011 were low (12.6%) among the households who were remained in better-off HEC in 2005 and 2011 as compared to other households (16.%) who remained in poor condition in both years. The members of household who were not sick with ASM in 2005 and fell sick with ASM in 2011 was higher (15.9%; P<0.05) among the households whose household environment has been turned downwards into poor condition in 2011 from better-off condition in 2005 as compared to other households (14.5%; P<0.05) whose household environment has been transformed into better-off condition in 2011 from poor in 2005.

**Table 6. Multivariate linear regression on percent sick in any short term morbidity members of households by change in household environment condition and socio-economic variables.**

| Change in Household Environment | (β coefficient) | P- value | 95%CI of β | |
|---|---|---|---|---|
| Remained in same HEC in 2005 & 2011 [R] | | | | |
| Poor HEC in 2005 and non-poor HEC in 2011 | -0.051 | <0.001 | [-0.086 | -0.016] |
| Non-poor HEC in 2005 and poor HEC in 2011 | 0.008 | <0.66 | [-0.027 | 0.042] |
| Non-poor HEC in in 2005 & 2011 | -0.060 | <0.001 | [-0.089 | -0.031] |
| **Place of residence** | | | | |
| Rural[R] | | | | |
| Urban | -0.015 | <0.26 | [-0.040 | 0.011] |
| **Type of house** | | | | |
| Kucha[R] | | | | |
| Pucca | -0.014 | <0.25 | [-0.039 | 0.010] |
| **Occupation of the head of the household** | | | | |
| No-occupation[R] | | | | |
| Primary | 0.002 | <0.91 | [-0.033 | 0.037] |
| Secondary | 0.009 | <0.68 | [-0.035 | 0.054] |
| Tertiary | 0.003 | <0.89 | [-0.039 | 0.045] |
| **Education of the head of the household** | | | | |
| Illiterate[R] | | | | |
| Primary | 0.004 | <0.78 | [-0.026 | 0.035] |
| Secondary | 0.021 | <0.12 | [-0.006 | 0.048] |
| Higher | 0.047 | <0.15 | [-0.017 | 0.110] |
| **Economic status** | | | | |
| Poor[R] | | | | |
| Non-poor | 0.089 | <0.001 | [0.065 | 0.113] |
| **Caste** | | | | |
| General[R] | | | | |
| OBC | -0.012 | <0.32 | [-0.037 | 0.012] |
| SC | -0.011 | 0.47 | [-0.039 | 0.018] |
| ST | -0.079 | 0.001 | [-0.122 | -0.037] |
| **Religion** | | | | |
| Hindu[R] | | | | |
| Muslim | -0.104 | <0.00 | [-0.135 | -0.074] |
| Christian | 0.039 | <0.05 | [-0.001 | 0.079] |
| **Region** | | | | |
| North[R] | | | | |
| Central | 0.147 | <0.001 | [0.117 | 0.176] |
| East | 0.053 | <0.001 | [0.022 | 0.084] |
| Northeast | 0.024 | <0.48 | [-0.041 | 0.089] |
| West | -0.054 | <0.001 | [-0.091 | -0.018] |
| South | 0.118 | <0.001 | [0.088 | 0.148] |
| Constant | 3.31 | <0.001 | [3.256 | 3.356] |
| Total number of observations | 15858.00 | | | |
| R-squared | 0.02 | | | |
| Adjusted R-squared | 0.02 | | | |
| Prob > F = 0.0000 | | | | |

[R] Stands for Reference category. CI stands for Confidence Interval.

**Table 7. Multinomial logistic regression analysis: Adjusted percentage of change in any short term morbidity by change in household environment condition and socioeconomic variables.**

| Background variables | No ASM in 2005 & 2011[®] | No ASM in 2005 & ASM in 2011 | ASM in 2005 & no ASM in 2011 | ASM in 2005 & 2011 |
|---|---|---|---|---|
| **Change in household environment condition** | | | | |
| Non-poor HEC in 2005 & 2011[®] | 75.7 | 12.6 | 9.0 | 2.7 |
| Poor HEC in 2005 & 2011 | 67.7 | 16.0 | 11.4** | 4.9** |
| Poor HEC in 2005 and non-poor HEC in 2011 | 71.1 | 14.5** | 11.0** | 3.3 |
| Non-poor HEC in 2005 and poor in HEC 2011 | 68.6 | 15.9** | 11.6*** | 3.9*** |
| **Place of residence** | | | | |
| Rural[®] | 70.7 | 14.8 | 10.7 | 3.8 |
| Urban | 76.8 | 12.0** | 8.6 | 2.6 |
| **Type of house** | | | | |
| Kucha[®] | 70.1 | 14.9 | 11.0 | 4.1 |
| Pucca | 76.7 | 12.4** | 8.5*** | 2.4*** |
| **Occupation of the head of the household** | | | | |
| Primary[®] | 71.7 | 14.5 | 10.3 | 3.5 |
| Secondary | 72.8 | 13.1 | 10.1 | 4.0 |
| Tertiary | 74.6 | 13.3 | 9.2 | 2.8 |
| No-Occupatiion | 70.9 | 13.8 | 11.0 | 4.3** |
| **Education of the head of the household** | | | | |
| Illiterate[®] | 71.5 | 14.6 | 10.2 | 3.7 |
| Primary | 72.6 | 13.7 | 10.4 | 3.3 |
| Secondory | 73.6 | 13.1 | 10.1 | 3.2 |
| Higher | 77.8 | 10.5*** | 8.3 | 3.4 |
| **Economic status** | | | | |
| Poor[®] | 72.8 | 15.3 | 9.1 | 2.8 |
| Non poor | 72.1 | 13.7 | 10.5*** | 3.7*** |
| **Social group** | | | | |
| General[®] | 73.5 | 13.4 | 9.9 | 3.2 |
| OBC | 71.9 | 14.7 | 9.9 | 3.6 |
| SC | 69.7 | 14.9 | 11.3 | 4.0 |
| ST | 76.9 | 10.9*** | 9.6 | 2.6** |
| **Religion** | | | | |
| Hindu[®] | 72.1 | 14.1 | 10.2 | 3.6 |
| Muslim | 71.4 | 14.8 | 10.0 | 3.7 |
| Christian | 76.0 | 12.1 | 9.5 | 2.3 |
| **Region** | | | | |
| North[®] | 74.2 | 13.6 | 9.2 | 2.9 |
| Central | 62.7 | 20.9*** | 11.1*** | 5.4*** |
| East | 68.2 | 13.6 | 13.2*** | 5.0*** |
| Northeast | 80.3 | 10.5*** | 8.0 | 1.2*** |
| West | 79.4 | 9.2*** | 9.3 | 2.1** |
| South | 77.9 | 12.3*** | 7.9*** | 1.9*** |
| Total | 72.2 | 14.1 | 10.2 | 3.5*** |
| Number of obs | 34131 | | | |
| LR chi2 | 1118.83 | | | |
| Prob > chi2 | 0.000 | | | |
| Pseudo R2 | 0.0195 | | | |

® Stands for Reference category. CI stands for Confidence Interval. P<0.1, P<0.05 and P<0.01.

The members of households who were not sick in 2005 and fell in sick in 2011 were low among the urban (12%; P<0.05) households as compared to rural (14.8%) counterparts. Similarly, the members of households who were not sick in 2005 and fell sick in 2011 was low among pucca (12.4%; P<0.05) households as compared to Kucha households (15%). The household members who were not sick in 2005 and get sick with ASM in 2011 is lower (10.5%; P<0.01) among households with educated household head as compared to households illiterate head (14.6%). The members who were not sick in 2005 and sick with ASM in 2011 have shown lower (10.9%; P<0.01) morbidity among STs as compared to General castes (13.4%). The lower morbidity among the ST households is probably due to under-reporting due to lower utilization of health care services because availability and accessibility of health care services were low in the place as forest where they live far away from general population living place in the rural area. The significant regional differences in morbidity is evident. The household members who were not sick in 2005 and sick with ASM in 2011 is higher in Central region (20.9%; P<0.01) as compared to the Northern region (13.6%). While the members of household who were not sick in 2005 and sick with ASM in 2011 is lower in Western (9.2%; P<0.01) followed by Northeast (10.5%; P<0.01) and Southern (12.3%; P<0.01) region households as compared to Northern region (13.6%).

The morbidity is remained lower for pucca household members (2.4%; P<0.01) in comparison with kacha households (4.1%). The central (5.4%; P<0.01) and eastern region (5.0%; P<0.01) households have shown higher morbidity as compared to northern region households (2.9%).

## Discussion

The HEC is an important predictor of morbidity (diarrhea, fever, cough and Acute Respiratory Infection). In this study, we have assessed the prevalence of members of households fell sick due to ASM by the change in HEC with various socio-economic characteristics in India. Also, the impact of change in HEC on morbidity after controlling for various socioeconomic characteristics of the households. We found that an average percentage of members of household fell sick due to ASM was lower among the households whose household environment was remained in better off condition in both years in 2005 and 2011 as compared to other households who have lived in poor condition in both years in 2005 and 2011. Similarly, the results from multivariate linear regression analysis shows that the percentage of household members fell sick due to ASM is significantly lower among the household whose HEC has changed from poor to clean environment in 2011 as compared to the households who remained in poor environment condition in both periods in 2005 and 2011 after controlling for a range of socio-economic characteristics of the households.

Decrease in morbidity of households members as a result of improvement in HEC from poor to non-poor condition during 2005–2011 period, may be due to the implementation of various government schemes such as National Rural Drinking Water Programme, Total Sanitation Programme, Household Clean Energy Programme, and National Rural Health Mission. Though all these Schemes may have laid the foundation for improving the HEC (HEC), they may not have fully achieved their objective due to poor operational management at ground level [45, 46].

Evidence from multinomial logistic regression analysis shows that the percentage of household members fell sick with ASM remained higher among the households who have lived in poor environment condition in both years in comparison to other households who were remained in better-off HEC in both years. The poor HEC persists due to open defecation, unimproved source of drinking water, un-hygienic practices like not washing hand with soap after defecating, not cleaning toilet properly, cooking in unhygienic place and combustion of

biomass for cooking. It further leads to an increase in the risk of morbidity among household members. Our finding is consistent with other studies [30–36, 47–53]. The unhygienic household environment causes morbidity among household members which leads to absenteeism from work thereby reduces the marginal productivity of the workers and causes decline in household income [42, 54].

While the household members fell in sick due to ASM has increased among the households whose environment condition has turned downwards into worse-off condition in 2011 from better-off condition in 2005. The increase in morbidity in the households whose environment condition has changed from better off to worse-off condition may be due to natural calamities like droughts and floods, leads to more use of unimproved source of water, cause no use of toilet facilities in 2011 even though they had water and sanitation facilities in the base period 2005. The natural calamities also push the household into poverty which cause more dependency on biomass fuel. Further, the studies have also reported that many Indian households do not use sanitation facilities and clean cooking fuel over the long period even these facilities have been provided earlier by government [20] that leads to poor HEC in later years.

Another important finding emerges in this study is that the morbidity among Muslim households is lower compared to Hindus. The lower morbidity among Muslims household members than the Hindus is mainly because they have better-off HEC such as pucca house, improved sanitation facilities, improved cooking fuel as compared to Hindus. The Muslims are less likely to defecate in open than the Hindu households even though Hindus are relatively economically advanced than the Muslims [55]. The household members living in pucca houses are less likely to have morbidity as compared to kucha households as the pucca houses have better off HEC than the kacha houses (houses made with poor quality materials for floor, roof and wall). This evidence is consistent with other studies that pucca (improved house as with cement and bricks) households adopted more improved toilet facility than kucha households [25].

## Conclusion

It is evident that the members of the households who were not sick in 2005 and fell sick in 2011 were lower among those living in urban households, pucca houses and with higher educated household head as compared to their counterparts in rural households, kucha houses and and illiterate household head. The household members living in Central and Eastern region have higher morbidity than other regions. The higher morbidity among rural households, kucha houses and households with illiterate household heads may be due to lack of awareness about clean household environment, poor WASH infrastructure, and non-usability of WASH facilities. Based on the findings of this study, we suggest that the ongoing programs launched during 2014–16 such as 'Swatch Bharat Abhiyan' (Clean India Mission), 'Pradhan Mantri Ujjwala Yojana' (Prime Mnister's Clean India Energy Programme) and 'Housing for All by 2020' need to concentrate among all these categories of households, especially in the Central and Eastern regions. It is essential to create health awareness among the vulnerable sections of the population to use the WASH facilities. Building a latrine alone would not make much sense until establishing a pucca house and providing access to running water facility. Building modern sanitation infrastructure is gaining importance to eradicate a wide range of morbidities in India. Lack of awareness about the association between HEC and health may be the reasons for higher proportion of household members who were living in poor HEC in both periods and the higher proportion of household members fell sick in ASM in the states like Bihar, Chhattisgarh, and Uttar Pradesh. These three states together houses of India's population as per 2011 census. These states need special focus by the ongoing programs. Those who are living in a clean environment are less likely to fall in sick with ASM. It has been observed

that without a healthy household environment, we cannot create a healthy population. Therefore, it is necessary to create a healthy HECs to prevent infectious diseases. The healthy population is an important component of human capital. Therefore, good health without any morbidity increases the working ability thereby increases the marginal productivity of the population. To create a healthy population, it is essential to build modern WASH facilities along with changing the behavior of the people to adopt improved sanitation and hygiene facilities through the behavior change communication strategies and information and communication technology.

## Author Contributions

**Conceptualization:** N. Brahmanandam, R. Nagarajan.

**Data curation:** N. Brahmanandam.

**Formal analysis:** N. Brahmanandam.

**Methodology:** N. Brahmanandam.

**Writing – original draft:** N. Brahmanandam.

**Writing – review & editing:** N. Brahmanandam, R. Nagarajan.

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
