## [Decision Letter · Decision Letter 0]

3 Dec 2020

PONE-D-20-35348

Impact of Change in household environment condition on morbidity in India: Evidence from Longitudinal Data

PLOS ONE

Dear Dr. Nuvula,

Thank you for submitting your manuscript to PLOS ONE. After careful consideration, we feel that it has merit but does not fully meet PLOS ONE’s publication criteria as it currently stands. Therefore, we invite you to submit a revised version of the manuscript that addresses the points raised during the review process.

 All SEVEN reviewers and I feel that this paper has merit for publication in PLOS with a minor revision. The one overwhelming suggestion that emerged from all seven reviewers is that your paper is technically sound and rigorous but you need to thoroughly work on the language of the paper. Kindly get your paper edited by professional language editors.

We look forward to receiving your revised manuscript.

Kind regards,

Srinivas Goli, Ph.D.

Academic Editor

PLOS ONE

Additional Editor Comments:

All SEVEN reviewers and I feel that this paper has merit for publication in PLOS with a minor revision. The one overwhelming suggestion that emerged from all seven reviewers is that your paper is technically sound and rigorous but you need to thoroughly work on the language of the paper. Kindly get your paper edited by professional language editors.

Journal Requirements:

Reviewers' comments:

Reviewer's Responses to Questions

**Comments to the Author**

1. Is the manuscript technically sound, and do the data support the conclusions?

Reviewer #1: Yes

Reviewer #2: Yes

Reviewer #3: Yes

Reviewer #4: Yes

Reviewer #5: Yes

Reviewer #6: Partly

Reviewer #7: Yes

2. Has the statistical analysis been performed appropriately and rigorously? 

Reviewer #1: Yes

Reviewer #2: Yes

Reviewer #3: Yes

Reviewer #4: Yes

Reviewer #5: Yes

Reviewer #6: Yes

Reviewer #7: Yes

3. Have the authors made all data underlying the findings in their manuscript fully available?

Reviewer #1: Yes

Reviewer #2: Yes

Reviewer #3: Yes

Reviewer #4: Yes

Reviewer #5: Yes

Reviewer #6: No

Reviewer #7: Yes

4. Is the manuscript presented in an intelligible fashion and written in standard English?

Reviewer #1: Yes

Reviewer #2: Yes

Reviewer #3: No

Reviewer #4: No

Reviewer #5: Yes

Reviewer #6: No

Reviewer #7: No

5. Review Comments to the Author

Reviewer #1: this is an important burning topic, effect of household environmental factor on morbidity. He has mentioned everything in the manuscript in planned way. Conclusion is clear and giving specific message to other researcher

Reviewer #2: This paper attempts to study the impact of change in household environmental conditions on morbidity using the evidence from longitudinal data. This paper brings an interesting analysis that will contribute to the body of literature. Although the authors could have applied some of the longitudinal statistical models such as difference-in-difference or propensity score matching, the current method is sufficient to carry out the empirical exercise. This paper may be published with minor revision. My observations on the paper are following.

1. The main purpose of the “Swatch Bharat Mission” was to make India open defecation free. On 2nd October 2019, the Prime Minister of India has declared that India is now open defecation free. In this context, one should bring evidence that open is still being practiced. Otherwise, sentences on “Swatch Bharat Mission” may be dropped.

2. Tables: the p-values should not be 0.00 or 0.000. It should be <0.001. Some of the tables also miss the notes for star marks (*).

3. Second last paragraph of the results: “The lower morbidity among the ST households is probably due to under-reporting.” Please provide a reference or concrete logical argument for this sentence.

4. Methods of abstract: “For empirical analysis purpose of this study…”. Please check grammar here and elsewhere in the manuscript. The sentence should be “the purpose of the empirical analysis”.

Reviewer #3: The selected issue taken up for the analysis is of contemporary relevance. The data analysis and econometric tools are appropriate. My specific comments are as follows:

1. The entire manuscript needs serious editing for the English language. It needs an in-depth review for grammar, typographical and punctuation errors right from abstract to conclusion including tables (for instance table no. 3). The whole manuscript should be revisited for language corrections. In the current form the paper cannot be accepted.

2. As the PLOS ONE is an international journal, the English version of any native schemes like Pradhan Mantri Awas Yojna or other such schemes should also be given. For example, the English version would be Prime Minister Housing Scheme.

3. The authors have taken two outcome variables – first is the percentage of household members fall sick due to any short-term morbidity past one month preceding the survey period. And, the second one is recoded variable. I didn’t find use of first outcome variable in the manuscript. If it has been used, the author may clarify the same.

4. The biggest disadvantage with the PCA for constructing composite index is that it gives normalised score which can not be used for over a period of time comparison (Means factor scores can be used for cross sectional analysis but not measuring progress over a period of time). Therefore, the classification of index score into four categories as (a) remaining poor in 2005 and 2011; (b) poor HEC in 2005 and non-poor in 2011; (c) non-poor HEC in 2005 and became poor in 2011; (d) remain non-poor HEC in 2005 and 2011 is not technically correct.

5. Authors claimed that the reliability, validity and suitability estimates are given in the table 1, but I didn’t find any such value in table 1.

6. Table 1 and table 2 are not properly referred in the manuscript. They are not explained also.

7. Several studies have noted through their detailed empirical analysis that Muslims are socio-economically poor as compared to Hindus. The study also recognised this point but somehow claims that HEC of Muslims is much better than Hindus. The findings are contradictory in nature.

Once again, I would like to emphasise that the paper needs a serious check for language corrections. After incorporating abovementioned suggestions, the paper may be accepted for publication.

Reviewer #4: The paper addresses a very vital question and thus should be considered for publishing after minor revisions. In current form however, the manuscript needs to be thoroughly checked for grammatical errors and sentence coherency.

Reviewer #5: Thank you for opportunity to review the manuscript entitled “Impact of Change in household environment condition on morbidity in India: Evidence from Longitudinal Data.” The study will be important contribution to understand the impact of household environment on overall human health. The study used nationwide data of two rounds to examine the changes in the household environment. The study objectives is clear and used robust statistical techniques to derive the results. However, I have few comments for authors to consider while revising the manuscript.

1. In the abstract correct the spelling Swatch Bharat Mission. It should be Swachh Bharat Mission.

2. Background: Instead of sayings links, authors can consider saying association or impact in subheadings.

3. Last paragraph of background, authors claiming first time this study investigating. I suggest authors to revise the statement to “to our knowledge to date this is first study investigating the casual relationship”

4. Authors should focus on SDGs context rather constantly refereeing to MDGs

5. In the discussion section authors must provide some links with Pradhan mantri ujjwala yojna or at least should speculate in regards to this scheme.

6. There are several typos in the text and some minor grammar issues. While revising the manuscript consider improving these issues.

Reviewer #6: This is an interesting paper that focuses on the ‘Impact of Change in household environment condition on morbidity in India: Evidence from Longitudinal Data’’ in the context of India. However, the author(s) should address the following weaknesses/comments before publication (please see the attachment for specific comments)

Reviewer #7: Review

The topic covered in the article is interesting and can be a good read for researchers. The manuscript attempts to explore the linkage between the household environment and health outcomes through panel data. Such attempts need to be encouraged to bring interesting research for readers. Though the topic covered is impressive, the manuscript, however, falls short in terms of argument building, grammar, and notably in the interpretation section. The paper requires substantial reworking. The interpretation section needs to be revised to improve flow of the argument and language coherence, besides correcting grammatical errors and spelling. An immediate suggestion would be to get this manuscript copy-edited from some professional.

The empirical analyses carried out in this manuscript are standard, and the results are elucidative. However, the results are poorly written. Some inferences drawn seem to be blanket statements, which need to be more nuanced. It is also preferable that the findings are discussed in the light of existing literature, especially if the results do not corroborate with what has been established earlier such as “The lower morbidity among the ST households is probably due to under-reporting” and “The morbidity has increased in the household … due to natural calamities like droughts and floods, leads to more use of unimproved source of water, cause no use of toilet facilities in 2011 even though they have water and sanitation facilities in the base period 2005” and “ The lowering in morbidity among Muslims than the Hindus is …improved cooking fuel as compared to Hindus”.

The authors need to explain as to what all variables constitute any short-term morbidity (ASM) in the study and how its share has changed over the period of time could be included in the Table-1. The independent variables such as occupation and education mentioned in Table 6 and 7 creates confusion as to whom it indicates the ‘Household head’ or ‘members in the household’. Similarly, for the same classification, authors have mentioned ‘Remain same in2005&2011’ and ‘Non-poor in 2005 2011’ at two distinct places. It is surprising to note that the authors have applied ANOVA test among the variables while its interpretation is completely missing in the study.

The same set of information is also available in the recent round of the National Family Health Survey (NFHS-4), and NSS 76th round wherefrom authors can assess the causality between HEC and ASM. Therefore, the authors must explain the importance of using IHDS for assessing the causal relations between HEC and ASM. Also, concluding remark on the various ongoing flagship Programme could be appealing if given in the light of findings from the study.

It is recommended that authors consider reworking on the result, discussion and conclusion sections and resubmit the manuscript for further evaluation and necessary actions.

Best Wishes

6. PLOS authors have the option to publish the peer review history of their article (what does this mean?). If published, this will include your full peer review and any attached files.

Reviewer #1: No

Reviewer #2: No

Reviewer #3: **Yes: **Nagendra Kumar Maurya

Reviewer #4: No

Reviewer #5: No

Reviewer #6: No

Reviewer #7: No

---

## [Author Response · Author response to Decision Letter 0]

19 Jan 2021

Point by Point Replay to Reviewer’s Comments

Manuscript ID No- PONE-D-20-35348: Impact of Change in household environment condition on morbidity in India: Evidence from Longitudinal Data 

Date-17-01-2020.

To,

Editor-in-Chief

PLOSE ONE

Thank you for giving us the opportunity to revise our Manuscript titled “Impact of Change in household environment condition on morbidity in India: Evidence from Longitudinal Data” (ID No- PONE-D-20-35348). My co-authors and I are grateful to you and the reviewers for insightful and constructive suggestions. These valuable positive suggestions and comments have helped us improve our manuscript substantially. We believed that the revised version of the manuscript address the comments raised by reviewers. The summary of our revisions are organized in the order of your comments below.

Reviewer –I

Comment -1: This is an important burning topic, effect of household environmental factor on morbidity. He has mentioned everything in the manuscript in planned way. Conclusion is clear and giving specific message to other researcher.

Author reply-I: Thank the referee for positive comment

Reviewer –II

Comment -II: Tables: the p-values should not be 0.00 or 0.000. It should be <0.001. Some of the tables also miss the notes for star marks (*).

Authors’ reply-II: The suggestion is well taken and the same is incorporated in the manuscript

Comment -1II: Second last paragraph of the results: “The lower morbidity among the ST households is probably due to under-reporting.” Please provide a reference or concrete logical argument for this sentence.

Authors’ reply-III: Normally the poorer sections of the population don’t consider/report the minor ailments as serious ones. People who are highly educated, richest and residing in developed regions report higher morbidity than their counterparts (less educated, poorer and residing in under developed regions). The suggestion is well taken and we have given concret logical argument for this statement in the manuscript.

Reviewer –III

Comment –I : As the PLOS ONE is an international journal, the English version of any native schemes like Pradhan Mantri Awas Yojna or other such schemes should also be given. For example, the English version would be Prime Minister Housing Scheme.

Authors’ reply-I: The suggestion is well taken and it is incorporated in the revised manuscript.

Comment -II: The authors have taken two outcome variables – first is the percentage of household members fall sick due to any short-term morbidity past one month preceding the survey period. And, the second one is recoded variable. I didn’t find use of first outcome variable in the manuscript. If it has been used, the author may clarify the same.

Authors’ reply-II: The comment is well taken. The first outcome variable is the percentage of household members fell sick due to any short term morbidity is a continuous variable. The main purpose of first outcome variable is to show the average percentage of household members fell sick due to any short term morbidity with respect change in household environment condition in different socioeconomic characteristics of the households. Whereas, it is not possible to show average percentage of household members fell sick due to any short term morbidity by second outcome variable with respect to change in household environment condition because the outcome variable is in categorical nature.

Comment -III: The biggest disadvantage with the PCA for constructing composite index is that it gives normalized score which can not be used for over a period of time comparison (Means factor scores can be used for cross sectional analysis but not measuring progress over a period of time). Therefore, the classification of index score into four categories as (a) remaining poor in 2005 and 2011; (b) poor HEC in 2005 and non-poor in 2011; (c) non-poor HEC in 2005 and became poor in 2011; (d) remain non-poor HEC in 2005 and 2011 is not technically correct.

Authors’ Response-III: IHDS is a panel data which follows-up the same households, thus, our index doesn’t suffer from over a period of time comparison. 

Comment -V: Authors claimed that the reliability, validity and suitability estimates are given in the table 1, but I didn’t find any such value in table 1.

Authors’ reply-V: The comment is well taken. The reliability and validity was estimated for 18 household environment condition variables in table-2. The reliability and validity’s values are provided at the bottom of the table-2 for round-1 and round-2 of 18 variables of household environment condition. 

 Comment -VI: Table 1 and table 2 are not properly referred in the manuscript. They are not explained also.

Author replay-VI: The comment is well taken. Table 1 gives the description of the data and table 2 gives the variables used for the construction of HEC index. Normally these tables are not needed to be explained in detail in the manuscript.

Comment -VII: Several studies have noted through their detailed empirical analysis that Muslims are socio-economically poor as compared to Hindus. The study also recognised this point but somehow claims that HEC of Muslims is much better than Hindus. The findings are contradictory in nature. 

Authors’ reply-VII: Thank you for careful observation: Even though Muslim households are socioeconomically poor, their household environment condition is better-off than the Hindu households in terms of improved sanitation facilities and hygienic conditions as mentioned by previous studies( Coffey and Spears 2017; Coffey 2014). IHDS data of two rounds show that the Muslim household have more improved household environment condition than the Hindu households in India which I have analysed and submitted the paper to another journal for the consideration.

Reviewers –IV, V and VII

Comments-I: Reviewers IV, V and VII have suggested that the paper addresses a very vital question and thus should be considered for publishing after minor revisions. In current form however, the manuscript needs to be thoroughly checked for grammatical errors and sentence coherency. The entire manuscript needs serious editing for the English language. It needs an in-depth review for grammar, typographical and punctuation errors right from abstract to conclusion including tables (for instance table no. 3). The whole manuscript should be revisited for language corrections. 

Authors’ reply-I. Thanks for the reviewers for their suggestion to improve the English language. We have improved the manuscript with respect to the English language.

Comment-II: In the abstract correct the spelling Swatch Bharat Mission. It should be Swachh Bharat Mission. The comment Background: Instead of sayings links, authors can consider saying association or impact in subheadings. The Last paragraph of background, authors claiming first time this study investigating. I suggest authors to revise the statement to “to our knowledge to date this is first study investigating the casual relationship”. The authors stated that: For the first time, this study investigates the causal relationships between household living environment and prevalence of short-term morbidities using data from the survey that followed the same households in 2004-05 and 2011-12. However, it is important to add information whether it is a first-time effort for the global context or in the context of India?

Author replay-II: Thank you for positive suggestions for improvement of the manuscript. We have incorporated the suggestions in the manuscript. We have revised the statement mentioning that “…… to jour knowledge to date this is the first study investigate the causal relation between any short term morbidity (Diarrhea, fever and cough) and household environment condition in the context of India”.

Comment-III: The study should explain the justification of using linear and multinomial logistic regressions (justification of using separate models) and of selecting the predictors precisely.

Author replay-III: The comment is well taken. The multivariate linear regression was used to assess the impact of change in household environment condition on morbidity. The outcome variable is percentage of household members fall sick due to any short term morbidity is a continuous variable. It shows the percentage of household members fell sick due to any short term morbidity is declined or increased/lower or higher with respect to change in household environment condition. It will not show the transition nature or change with respect to transition or change in household environment condition. Therefore, the multinomial logistic regression is used to assess the change in morbidity by change in household environment condition. In this context, the outcome variable shows that change or transition nature of morbidity is multinomial categories as remained same morbidity in both years in 2005 and 2011, higher morbidity in 2005 but lower morbidity in 2011, the lower morbidity in 2005 but higher morbidity in 2011 and remained same in both years in 2005 and 2011, with respect change in household environment condition. The morbidity variable is shown as transition in nature in this paper. We have incorporated the justification for using linear and multinomial logistic regressions in the paper.

Comment-IV: Data (in %) for the sentences ‘’The average percentage of household members fell sick due to ASM was declined among the households who have experienced a better change in household environment conditions in 2011 compared to 2005’ and ‘Whereas average percentage of household members fall sick due to morbidity has been increased among the households who were lived in a clean environment condition in 2005 but they moved into poor household environment conditions in 2011’’ are needed.

Author replay-IV: Wherever necessary, we have added the percentages in the sentences or in parentheses as per the suggestion of the referee. 

Comment-V: The authors should incorporate references in several places of the discussion section such as: The sentence in 2nd paragraph (it is also suggested to split): ---due to implementation of the government schemes such as National Rural Drinking Water Programme, Total Sanitation Programme, Household Clean Energy Programme, and National Rural Health Mission, All these Schemes laid the foundation to improve household environment condition (HEC) under United Progressive Alliance (UPA) government, though, all these schemes successfully not achieved their aims due to poor operational management at ground level. 

Author replay-V: We have included the reference as per the suggestion of the referee. 

Comment-VI: Add description *** at the end of each table, where appropriate

Authors’ reply-VI: We have added the description as per the suggestion of the referee.

Comment-VII: To what extend the linear regression model is suitable to explain the effects of predictors on outcome measure when the adjusted R-square is 0.02?

Authors’ Response-VII: Adjusted R-square is too small because of controlling for too many variable in the model. However, lower adjusted r-square doesn’t qualify to completely reject the model. 

Comment-VIII: It is surprising to note that the authors have applied ANOVA test among the variables while its interpretation is completely missing in the study.

Author replay-VIII: The comment is well taken and the same is incorporated in the revised manuscript.

---

## [Decision Letter · Decision Letter 1]

8 Feb 2021

Impact of Change in household environment condition on morbidity in India: Evidence from Longitudinal Data

PONE-D-20-35348R1

Dear Dr. Nuvula,

We’re pleased to inform you that your manuscript has been judged scientifically suitable for publication and will be formally accepted for publication once it meets all outstanding technical requirements.

Kind regards,

Srinivas Goli, Ph.D.

Academic Editor

PLOS ONE

Additional Editor Comments (optional):

Authors addressed all the comments raised in the previous version and the paper can be accepted for the publication.

Reviewers' comments:

Reviewer's Responses to Questions

**Comments to the Author**

1. If the authors have adequately addressed your comments raised in a previous round of review and you feel that this manuscript is now acceptable for publication, you may indicate that here to bypass the “Comments to the Author” section, enter your conflict of interest statement in the “Confidential to Editor” section, and submit your "Accept" recommendation.

Reviewer #2: All comments have been addressed

Reviewer #5: All comments have been addressed

2. Is the manuscript technically sound, and do the data support the conclusions?

Reviewer #2: Yes

Reviewer #5: Yes

3. Has the statistical analysis been performed appropriately and rigorously? 

Reviewer #2: Yes

Reviewer #5: Yes

4. Have the authors made all data underlying the findings in their manuscript fully available?

Reviewer #2: (No Response)

Reviewer #5: Yes

5. Is the manuscript presented in an intelligible fashion and written in standard English?

Reviewer #2: (No Response)

Reviewer #5: Yes

6. Review Comments to the Author

Reviewer #2: The comments made by me have been carefully addressed. The paper in this current form may be accepted.

Reviewer #5: Authors addressed all the concerns raised during the first review comment this manuscript may be accepted for publication.

7. PLOS authors have the option to publish the peer review history of their article (what does this mean?). If published, this will include your full peer review and any attached files.

Reviewer #2: No

Reviewer #5: No

---

## [Editor Report · Acceptance letter]

15 Feb 2021

PONE-D-20-35348R1 

Impact of change in household environment condition on morbidity in India: Evidence from longitudinal data 

Dear Dr. .N:

I'm pleased to inform you that your manuscript has been deemed suitable for publication in PLOS ONE. Congratulations! Your manuscript is now with our production department. 

Kind regards, 

on behalf of

Dr. Srinivas Goli 

Academic Editor

PLOS ONE